# UK-based, multisite, prospective cohort study of small bowel obstruction in acute surgical services: National Audit of Small Bowel Obstruction (NASBO) protocol

Matthew J Lee,[1,2] Adele E Sayers,[2,3] Thomas M Drake,[2,4] Marianne Hollyman,[5,6] Mike Bradburn,[7] Daniel Hind,[7] Timothy R Wilson,[8,9] Nicola S Fearnhead,[9,10] on behalf of the NASBO Steering Group

## ABSTRACT

**Introduction** Small bowel obstruction (SBO) is a common indication for emergency laparotomy in the UK, which is associated with a 90-day mortality rate of 13%. There are currently no UK clinical guidelines for the management of this condition. The aim of this multicentre prospective cohort study is to describe the burden, variation in management and associated outcomes of SBO in the UK adult population.

**Methods and analysis** UK hospitals providing emergency general surgery are eligible to participate. This study has three components: (1) a clinical preference questionnaire to be completed by consultants providing emergency general surgical care to assesses preferences in diagnostics and therapeutic approaches, including laparoscopy and nutritional interventions; (2) site resource profile questionnaire to indicate ease of access to diagnostic services, operating theatres, nutritional support teams and postoperative support including intensive care; (3) prospective cohort study of all cases of SBO admitted during an 8-week period at participating trusts. Data on diagnostics, operative and nutritional interventions, and in-hospital mortality and morbidity will be captured, followed by data validation.

**Ethics and dissemination** This will be conducted as a national audit of practice in conjunction with trainee research collaboratives, with support from patient representatives, surgeons, anaesthetists, gastroenterologists and a clinical trials unit. Site-specific reports will be provided to each participant site as well as an overall report to be disseminated through specialist societies. Results will be published in a formal project report endorsed by stakeholders, and in peer-reviewed scientific reports. Key findings will be debated at a focused national meeting with a view to quality improvement initiatives.

For numbered affiliations see end of article.

**Correspondence to**
Matthew J Lee;
m.j.lee@sheffield.ac.uk

## Strengths and limitations of this study

► This study will be the largest prospective assessment of the management of small bowel obstruction in adults in the UK.
► This study will highlight variation in resources and clinical practice, and assess the impact of variation on patient outcomes.
► The methodology limits data to easily measured key components of the treatment pathway that are routinely captured in patient notes.
► Accuracy of data collection will be assessed in a short post hoc validation exercise.
► Potential inclusion of all hospitals providing emergency general surgery will ensure that findings are broadly representative of UK practice.

## BACKGROUND

Mechanical small bowel obstruction (SBO) is a common presentation to emergency general surgery. Eleven and a half thousand patients in England and Wales underwent emergency laparotomy for SBO during the 12 months from April 2014 to March 2015.[1] This was associated with a 90-day mortality rate of 13%.[1] Similar findings have been noted in the USA.[2]

SBO has several aetiologies, including congenital or postoperative adhesions, abdominal wall hernia and malignancy. Plain film radiography or CT may be used to confirm the diagnosis and determine underlying cause. Depending on aetiology and comorbidities, patients may be selected for early surgical intervention or conservative management, typically with nasogastric decompression, urinary catheterisation and intravenous fluid therapy.[3 4] Around two-thirds of patients managed conservatively for adhesive SBO will settle, but the remainder will require surgery,[5] with a prolonging of the treatment pathway and time to gastrointestinal recovery (figure 1).

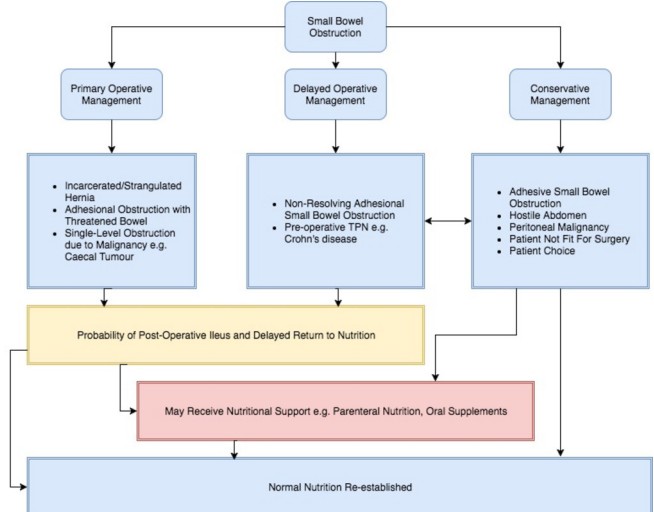

**Figure 1** Conceptual schematic of pathways in the management of small bowel obstruction, including typical diagnoses and nutritional outcomes. TPN, total parenteral nutrition.

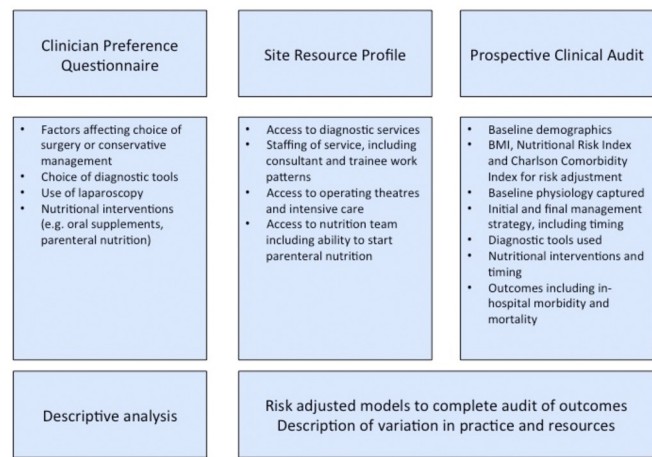

**Figure 2** Components of NASBO (National Audit of Small Bowel Obstruction) study, and how they are related. BMI, body mass index.

Guidelines already exist in the USA and Europe for the management of SBO.[3 4] The Royal College of Surgeons of England has described a pathway for the management of SBO, although this is presented in guidelines for the commissioning of emergency services, rather than clinical guidelines.[6] This advocates the use of early CT scanning, use of Gastrografin and timely intervention. Limited specific guidance leads to variation in the management of SBO across the UK.

Currently available data do not provide a national overview of the management of SBO: the National Emergency Laparotomy Audit (NELA) captures only the subset of patients who undergo surgery, meaning that we have no high-quality information on those managed conservatively and their outcomes.[1] As SBO accounts for half of emergency laparotomies, and likely many more conservatively managed patients, data to inform policy, quality improvement programmes and clinical trials are an audit priority.[7 8]

## AIM
The aim of this study is to describe the variation in management and outcomes of SBO in the UK.

Objectives of the study are to describe:
► variations in consultant practice in the management of SBO;
► variation in resources available to support the management of SBO;
► patient pathways and variation in the management of SBO;
► use of diagnostics in SBO (CT, plain film radiography);
► interventions used in SBO (operative intervention, therapeutic trial of water-soluble contrast agent);
► use of nutritional assessment tools and resulting nutritional interventions;
► rate of in-hospital mortality in patients treated for SBO;

► rates of 30-day readmission following treatment for SBO;
► rates of unplanned escalation to intensive care.

## METHODS
This project has three components: a survey of clinical practice, a site resource questionnaire and a prospective cohort study (figure 2). Site recruitment has been undertaken through specialty association conferences and electronic mailing, recruitment presentations at specialty meetings, through trainee research collaboratives and through professional contacts. All UK hospitals providing emergency general surgery are eligible to participate. This project has been registered with the Healthcare Quality Improvement Partnership.

### Survey of clinical practice
An anonymous survey of clinical practice has been prepared. This is to be completed only by consultant surgeons who provide emergency general surgery care—these clinicians are ultimately responsible for the inpatient management of this group and their preferences should influence care rather than other grades of doctor or other specialties. This captures basic demographic data including specialty and year of graduation. To contextualise clinical data, respondents are asked to indicate the impact of specific clinical factors on the selection of primary operative or conservative management (eg, multilevel obstruction due to disseminated malignancy, raised or normal inflammatory markers), the minimum investigations required for management, use of Gastrografin and use of laparoscopy. The survey also investigates preferences around nutritional support in SBO. Based on previous experience of surveying surgeons in areas with limited guidance, concerns have been expressed about providing responses out of line with the majority of the profession. In order to maximise returns, we decided to make this anonymous. This means that we cannot link back to institutions.

 Lee MJ, *et al. BMJ Open* 2017;**7**:e016796. doi:10.1136/bmjopen-2017-016796

### Site resource profile

The site resource profile is to be completed once for each participating site. This captures data on staffing levels, ease of access to diagnostics, theatres and nutritional support teams. This will indicate frequency of handovers of care and delays in access to diagnostics: these factors that may delay decision-making for these patients. Access to theatres, intensive care and nutritional support teams will indicate resource for implementing these decisions. The questionnaire also assesses availability of resources on weekdays, weekends and overnight.

### Prospective cohort study

Patients eligible for inclusion in the prospective cohort study must have met the following criteria:
▶ have been admitted from the emergency department or primary care to the acute surgery team or referred from an inpatient team to the emergency surgery team;
▶ a clinical diagnosis of SBO made by a specialty trainee year 3 or higher in general surgery.

These inclusion criteria are purposefully broad with the intention of capturing as many patients with SBO as possible.

Patients will be excluded if:
▶ they have undergone abdominal surgery within the same hospital admission prior to first symptoms of SBO;
▶ they are pregnant;
▶ they are under the age of 16 years old;
▶ they have large bowel obstruction (even when signs of SBO are present), for example, obstructing rectal tumour;
▶ they have length of stay <24 hours (discharged home).

Where the initial diagnosis changes, patients will be excluded retrospectively. Patients will be identified over an 8-week period. This period has been selected based on pilot data and NELA reports—to ensure a representative sample of cases and facilitate meaningful analysis, we set a target of 1500 cases. Extrapolation of numbers from a multisite pilot suggested that >2000 cases would be identified during a 2-week period, with an exclusion rate of around 20%. Consideration was also given to rotation of junior medical staff, who undertake the majority of data collection, and the period avoids most rotation dates. Data to be captured include basic demographics, comorbidities in the form of the Charlson comorbidity index[9] and usual place of residence (own home, residential home, nursing home) as a proxy for frailty (online supplementary appendix 1). Height and weight are captured to calculate body mass index (BMI) and nutritional risk index as risk adjustment tools.[10]

Data will be recorded on initial and final management strategies, baseline physiology, diagnostics and nutritional support strategies.

The primary outcome is in-hospital mortality. Secondary outcomes include in-hospital morbidity, length of stay and 30-day readmission.

Data will be uploaded to an encrypted and password-protected secure REDCap (Research Electronic Data Capture) server, hosted at the University of Sheffield.[11] No identifiable data are uploaded. Collaborators will keep a local 'key' spreadsheet linking REDCap identifiers to NHS or Hospital Numbers on their NHS network.

### Data validation

Only data sets with >95% data completeness will be accepted. Doctors at core trainee level or above, who were not involved in initial data collection, will act as independent assessors, reviewing data collected at a local centre. Overall independent assessors will validate a minimum of 10% of patient records, with a target of >95% case ascertainment and >90% data accuracy.

The number of identified patients having surgery during the audit period will be compared with those recorded in the NELA database for the same period. This will give an indication of how representative the data set is.

### Pilot

The survey has undergone pilot at two separate sites, with minor revisions after each round.

The prospective audit and site profile questionnaire have undergone a 2-week pilot across eight UK centres to confirm acceptability of definitions and usability of REDCap system.

### Anticipated recruitment

Based on NELA data for 2014–2015[1] and pilot work, we anticipate mean identification rates of three cases/week per centre. Across 100 centres, anticipated recruitment would be 2400 cases.

### Statistical analysis

Analysis will be performed by a statistician at the Clinical Trials Research Unit, University of Sheffield. Descriptive analysis will be performed to describe crude rates of mortality and morbidity, with subgroup analysis of primary operation, conservative management and failed conservative management. BMI, nutritional risk index[10] and Charlson comorbidity index[9] will be used for risk adjustment. Descriptive reporting of the use of diagnostics, operative approach and nutritional support in the treatment pathway will be performed, and association with outcomes recorded.

Variation in patient characteristics was taken into account during study design and will be taken into account during statistical analysis. Due to the expected heterogeneity across all patients, only clinically valid comparisons will be made according to the care pathways outlined in figure 1 (ie, initial operative management, successful conservative management or failed conservative management). During statistical analysis, multilevel modelling will allow differences across centres to be taken into account. Multilevel logistic regression models will be constructed using clinically plausible variables to identify predictors of mortality and morbidity following SBO.

Effects of predictor variables will be presented as OR, alongside the corresponding 95% CI. Sensitivity analyses stratified by number of cases per centre (in the case where hospitals have fewer than five cases) will be performed to assess and identify any changes to the direction and effect size which may be influenced by the inclusion of centres with few cases.

Data will be matched to site resource profiles to assess the relationship between resource availability and management practices.

### Ethics and governance

This project has been assessed by the Scientific Officer of the South East Scotland Research Ethics Service, who confirmed that the project did not require ethical approval. All sites must secure local audit approval prior to collecting data, and Information Governance or Caldicott approval prior to uploading data to REDCap. Caldicott approval for Scotland will be secured through a single central application.

### Funding

This project has received funding from the Bowel Disease Research Foundation, Association of Coloproctology of Great Britain and Ireland, Association of Surgeons of Great Britain & Ireland, Association of Upper Gastrointestinal Surgeons, British Association of Parenteral and Enteral Nutrition, British Association for Surgical Oncology, British Society of Gastroenterology, Royal College of Surgeons of England, Royal College of Surgeon of Edinburgh, NELA and Royal College of Anaesthetists.

### Authorship

All collaborators returning complete and validated data sets within the timelines will be eligible for collaborative authorship. This will be reported in line with the CRediT taxonomy.[12] We intend that each site has no more than four collaborators.

## DISCUSSION

SBO carries significant morbidity and mortality; however, most work on this topic has focused on specific diagnostic or therapeutic interventions, with little focus on how to address the associated high levels of mortality. The guidance from Eastern Association for the Surgery of Trauma, and World Society for Emergency Surgery offers extensive information on the use of CT scans to identify strangulation or 'high grade' SBO and the selection of patients for surgery (and operative approach), or conservative management.[3 4] This guidance does not substantially address other issues such as nutritional interventions, use of total parenteral nutrition or considerations in postoperative care.

This study will deliver the largest prospective assessment of the management of SBO in adults in the UK. Using clinical data on management of SBO, clinician management preferences and a local resource profile, we will report variation in management of this condition. These data will also permit early exploration of factors associated with variation in practice, and their relationship to outcomes. This study will also provide preliminary data on interventions used in SBO to re-establish feeding. Other studies in the field have focused only on specific areas of SBO management and to our knowledge, there are very limited data with regard to how nutrition is handled. The central aim of the NASBO (National Audit of Small Bowel Obstruction) project is to address this by delivering high-quality data across multiple centres.

This project uses multiple methods to accumulate data including surveys and clinical data collection. Surveys have been carefully designed and piloted to ensure validity and clarity of questions.

The snapshot clinical data capture has been designed to capture key components of the SBO pathway. While it captures several key nodes of clinical practice, it does not report on the use of nasogastric tubes or use of intravenous fluids. While these are commonly used, accurate data capture to describe them would require a significant amount of resource for what is likely to be highly granular data. If required, these factors could be explored in future studies delivered by the NASBO network. The treatment pathway and pathophysiology of SBO is complex and varied. This complexity, however, must be balanced with the ability to deliver high-quality useable data. This balance has been emphasised when designing the study and developing data collection tools.

Trainee research collaboratives have previously demonstrated the ability to deliver large multicentre studies.[13 14] This study differs in that it is the first time UK trainee research collaboratives have partnered with a number of specialty organisations and policymakers. The complexity of patient pathways and variation in clinical decision-making make SBO a prime target for intervention. Use and timing of CT, nutritional support and surgical intervention are all potentially costly interventions which are accompanied with risks to the patient. Therefore, it is imperative to generate a high-quality evidence base in a condition which carries a high mortality and morbidity rate. High-quality data on SBO will also allow appropriate assessment of the health economic impact of future interventions. Findings of this study will be used to inform development of clinical guidelines, quality indicators, and support development of clinical trials in the field.

We envisage this project will allow a network to be formed by clinicians who have an interest in improving outcomes following SBO. This network will permit the delivery of quality improvement projects and further, interventional research studies to be performed based on the results of the inaugural NASBO study.

**Author affiliations**
[1]Department of General Surgery, Sheffield Teaching Hospitals NHS Foundation Trust, Sheffield, South Yorkshire, UK
[2]South Yorkshire Surgical Research Group, Sheffield, UK
[3]Department of General Surgery, Mid-Yorkshire NHS Trust, Wakefield, UK

[4]Department of Clinical Surgery, University of Edinburgh, Edinburgh, UK
[5]Department of General Surgery, North Bristol NHS Trust, Bristol, UK
[6]Severn and Peninsula Audit and Research Collaborative, Bristol, UK
[7]Clinical Trials Research Unit, School of Health and Related Research, Sheffield, UK
[8]Department of General Surgery, Doncaster and Bassetlaw Teaching Hospitals NHS Foundation Trust, Doncaster, UK
[9]Association of Coloproctology of Great Britain and Ireland, London, UK
[10]Department of Colorectal Surgery, Addenbrooke's Hospital, Cambridge University Hospitals NHS Foundation Trust, Cambridge, Cambridgeshire, UK

**Twitter** @NASBO2017

**Collaborators** John Abercrombie, Austin G Acheson, Derek Alderson, Iain Anderson, Simon Bach, Michael Davies, Zaed Hamady, John Hartley, John Northover, Christopher Lewis, Paul Marriott, Nicholas Maynard, Malcolm McFall, Aravinth Muragananthan, David Murray, Pritam Singh, Gillian Tierney, Azmina Verjee, Ciaran Walsh and Jonathan Wild.

**Contributors** All authors and collaborators contributed to the development of the protocol for the project. Main drafts of text and revisions were undertaken by MJL, AES, TMD, MH, MB, DH, NSF and TRW. All authors including collaborators have reviewed and approved the manuscript.

**Funding** This project has been funded by the Bowel Disease Research Foundation, Association of Coloproctology of Great Britain and Ireland, Association of Surgeons of Great Britain and Ireland, Association of Upper Gastrointestinal Surgeons, British Association of Parenteral and Enteral Nutrition, British Association for Surgical Oncology, British Society of Gastroenterology, Royal College of Surgeons of England, Royal College of Surgeons of Edinburgh, National Emergency Laparotomy Audit and Royal College of Anaesthetists.

**Competing interests** None declared.

**Provenance and peer review** Not commissioned; externally peer reviewed.

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
