## [Reviewer comments · BMJ Open]

ARTICLE DETAILS

TITLE (PROVISIONAL)	A UK based, multi-site, prospective cohort study of Small Bowel Obstruction in acute surgical services: National Audit of Small Bowel Obstruction (NASBO) protocol.
AUTHORS	Lee, Matthew; Sayers, Adele; Drake, Thomas; Hollyman, Marianne; Bradburn, Mike; Hind, Daniel; Wilson, Timothy; Fearnhead, Nicola; Steering Group, NASBO

VERSION 1 – REVIEW

REVIEWER	Hajo Zeeb Leibniz-Institute for Prevention Research and Epidemiology-BIPS, Bremen, Germany
REVIEW RETURNED	04-May-2017

GENERAL COMMENTS	This is a protocol for a multi-centre prospective cohort study on small bowel obstruction among adults in the UK. All UK hospitals providing surgical emergency care are eligible to participate. P 7 L 10 – specify the 12 months (say from 4/2014-3/2015), since there are more than 12 months in the period 2014-2015. Delete “IN” L12: this was associated with indication ? makes no sense L 55 greater than what? P9 whereas outcomes are mentioned as aim, no outcome appears among the objectives. The authors surely want to describe in-hospital mortality, perhaps 30 - days mortality as well. On page 12, this is specified, but not here under the objectives. From the questionnaire in the appendix, it seems that cause of death will not be abstracted, are such data available? They might be interesting. P10 L27 the practice survey is anonymous, but is there a chance to link it to the institution – at least for the researchers? P9 L53 do you plan to assess resources distribution over week-days and week-ends separately? P11 inclusion criteria –are there cases where the initial diagnosis is not maintained? Will they be excluded retrospectively? What is the rationale for the eight- weeks period? Did you do any sample size calculations in advance, is this a purely pragmatic choice? Please explain. P11 L55: what if the patient with SBO dies within 24 hours, they should not be excluded? Perhaps this does not happen, the authors probably have some data. P 13 L48 crude instead of raw rates. Overall, the statistics are descriptive. Risk adjustment is mentioned, using which approach? I would expect some multivariate modelling of rates. Please specify.
---

	Are there no plans, even exploratory, to assess potential predictors of mortality or morbidity? Given the expected outcome frequency, this may be worth looking at. P15 L10 you may have very small centers with a handful of cases over the eight weeks. I recommend the authors plan for a minimum number of cases, or for a grouping of hospitals according to resources (or sth. along those lines) for a meaningful analysis. P 16 Discussion – this section provides some good arguments why this study is considered to be useful, what it adds and how it differs from earlier work. What I was missing was a statement whether the audit will be used for guideline development, since the authors point out the lack of clinical guidelines in the introduction. General comment: a) it may be worthwhile to consider a 90-days follow-up as well, for a more specific description of SBO-related mortality that can be compared to available literature. b) trial registration - if there are options to register non-clinical trials, the trial should be registered.
--	---

REVIEWER	Dion Morton University of Birmingham England
REVIEW RETURNED	18-Jul-2017

GENERAL COMMENTS	Transition from non-surgical to surgical management is a key step in the emergency care pathway and central to the management of Small bowel obstruction. This study seeks to address this transition and how it might effect outcomes. The question is of international importance and the researchers should be congratulated for bringing together a multicentre multidisciplinary team to address this. Specifically the inclusion of trainees in recruitment and data collection is an essential step to deliver this prospective study in the emergency setting. I agree with the authors' point that this could form a valuable network of researchers into emergency surgical care. The integration of resource availability into the study, will also make this data more relevant and useful to international investigators wishing to make comparisons in the future. A few minor suggestions  1. P7, L51 - reference is made to existing guidelines - some assessment of the quality of the evidence will add value to the results. Guidelines are sometimes not taken up, if the evidence for them is weak. 2. P11,L24 - I am not clear what the inclusion criteria are for the clinician survey. Is it limited to recruiting clinicians/surgeons/A&E doctors/Trainees/Consultants? I wonder if this could be added? 3. P11, L24 - diagnosis of SBO. This is challenging, because in order to be inclusive, the criteria are broad. However this will necessarily mean that some centres will include patients that others would not and this could influence findings, particularly around outcomes. I wonder if the methodologists could describe how this baseline variation will be managed. I suspect that addressing this up front will give greater impact to the findings.
---

VERSION 1 – AUTHOR RESPONSE

Reviewer 1

This is a protocol for a multi-centre prospective cohort study on small bowel obstruction among adults in the UK. All UK hospitals providing surgical emergency care are eligible to participate.

P 7 L 10 – specify the 12 months (say from 4/2014-3/2015), since there are more than 12 months in the period 2014-2015. Delete “IN”

This has been amended.

L12: this was associated with indication ? makes no sense

This has been corrected (removed).

L 55 greater than what?

This was meant to indicate high levels of variation. It was a grammatical error and has been removed.

P9 whereas outcomes are mentioned as aim, no outcome appears among the objectives. The authors surely want to describe in-hospital mortality, perhaps 30 - days mortality as well. On page 12, this is specified, but not here under the objectives. From the questionnaire in the appendix, it seems that cause of death will not be abstracted, are such data available? They might be interesting.

We have clarified mortality, readmission, and unplanned escalation of care as objectives (page 9). We considered capturing cause of death, but decided against this for reasons of practicality. These include burden of data collection on local teams, and potential delays in confirmation of cause of death due to legal processes in cases where the patient has undergone an operation.

P10 L27 the practice survey is anonymous, but is there a chance to link it to the institution – at least for the researchers?

During the planning stage, we decided that we would make this completely anonymous. Based on previous experience of surveying surgeons in areas with limited guidance, concerns have been expressed about providing responses out of line with the majority of the profession. In order to maximise returns, we decided to keep this anonymous. This means that we cannot link back to institutions. This has been stated on page 10.

P9 L53 do you plan to assess resources distribution over week-days and week-ends separately?

The resource questionnaire assesses resource availability during weekdays, during weekends and overnight (p11).

P11 inclusion criteria –are there cases where the initial diagnosis is not maintained? Will they be excluded retrospectively? What is the rationale for the eight- weeks period? Did you do any sample size calculations in advance, is this a purely pragmatic choice? Please explain.

Yes, where the diagnosis changes from SBO, the patient can be excluded retrospectively. We have now explained how we selected the 8-week window based upon our pilot data.

P11 L55: what if the patient with SBO dies within 24 hours, they should not be excluded? Perhaps this

does not happen, the authors probably have some data.

If a patient dies within 24 hours of admission, they should still be included. The 24 hours length of stay refers to those who are discharged home alive only. This has been clarified.

P 13 L48 crude instead of raw rates. Overall, the statistics are descriptive. Risk adjustment is mentioned, using which approach? I would expect some multivariate modelling of rates. Please specify. Are there no plans, even exploratory, to assess potential predictors of mortality or morbidity? Given the expected outcome frequency, this may be worth looking at.

There is no existing model which covers both operative and non-operative management of small bowel obstruction. Multilevel logistic regression models will be constructed using clinically plausible variables to identify predictors of mortality and morbidity. Effects of predictor variables will be presented as odds ratios (OR), alongside the corresponding 95% confidence interval. We have now stated this in the statistical analysis section.

P15 L10 you may have very small centers with a handful of cases over the eight weeks. I recommend the authors plan for a minimum number of cases, or for a grouping of hospitals according to resources (or sth. along those lines) for a meaningful analysis.

This is a potential issue. We have avoided setting a minimum recruitment numbers as we are also interested in how the smallest/least busy units manage this condition. Sensitivity analyses stratified by number of cases per centre (in the case where hospitals have fewer than 5 cases) will be performed to assess identify any changes to the direction and effect size which may be influenced by the inclusion of centres with few cases. This is now stated in the statistics section.

P 16 Discussion – this section provides some good arguments why this study is considered to be useful, what it adds and how it differs from earlier work. What I was missing was a statement whether the audit will be used for guideline development, since the authors point out the lack of clinical guidelines in the introduction.

We hope that this study will inform clinical guideline development, quality indicators, and potentially a clinical trial.

General comment: a) it may be worthwhile to consider a 90-days follow-up as well, for a more specific description of SBO-related mortality that can be compared to available literature. b) trial registration - if there are options to register non-clinical trials, the trial should be registered.

For practical reasons within the confines of audit requirements and data-sharing, we have avoided patient identifiers and therefore cannot undertake linkage of data to administrative databases. We would undertake this in follow-up studies focused on higher volume centres. This study has been registered with the Healthcare Quality Improvement Project (HQIP) as a National Audit (No reference number available).

Reviewer 2:

Transition from non-surgical to surgical management is a key step in the emergency care pathway and central to the management of Small bowel obstruction.

This study seeks to address this transition and how it might effect outcomes.

The question is of international importance and the researchers should be congratulated for bringing

together a multicentre multidisciplinary team to address this. Specifically the inclusion of trainees in recruitment and data collection is an essential step to deliver this prospective study in the emergency setting. I agree with the authors' point that this could form a valuable network of researchers into emergency surgical care. The integration of resource availability into the study, will also make this data more relevant and useful to international investigators wishing to make comparisons in the future.

Thank you for these comments. The engagement of trainees is key to delivering this project, and future projects.

A few minor suggestions

1. P7, L51 - reference is made to existing guidelines - some assessment of the quality of the evidence will add value to the results. Guidelines are sometimes not taken up, if the evidence for them is weak.

We agree. We have already undertaken work outside the scope of this manuscript in assessing current available evidence for the management of small bowel obstruction.

2. P11,L24 - I am not clear what the inclusion criteria are for the clinician survey. Is it limited to recruiting clinicians/surgeons/A&E doctors/Trainees/Consultants? I wonder if this could be added?

The clinician survey is designed to be completed by consultant surgeons only, as they will have set preferences in their practice and have ultimate responsibility for the management of patients. This has been clarified at the start of the survey of clinical practice paragraph on page 10.

3. P11, L24 - diagnosis of SBO. This is challenging, because in order to be inclusive, the criteria are broad. However this will necessarily mean that some centres will include patients that others would not and this could influence findings, particularly around outcomes. I wonder if the methodologists could describe how this baseline variation will be managed. I suspect that addressing this up front will give greater impact to the findings.

This point is correct and we agree that this is a potential issue. We have addressed this in the statistical analysis section, describing use of multilevel regression and sensitivity analyses.

I hope that we have addressed the reviewers comments appropriately, and we would be happy to address any further questions or comments.

VERSION 2 – REVIEW

REVIEWER	Hajo Zeeb Leibniz-Institute for Prevention Research and Epidemiology-BIPS, Germany
REVIEW RETURNED	20-Aug-2017

GENERAL COMMENTS	The authors have done a good job in responding to and clarifying issues raised in my initial review. I understand the choice to do the audit without identifiers, but wonder if some kind of grouped identifier might still be used in order to be able to also give targeted feedback to particular institutions, perhaps on a (sub)-regional basis.
---